# Implicit 3D Reconstruction of Fine Details from Multi-View Images using Wavelet-based Geometric Prior

## Abstract

High-fidelity 3D reconstruction from images remains a fundamental challenge in computer vision. Implicit Signed Distance Field (SDF) models leverage photometric loss for isosurface reconstruction, while recent approaches, such as planar constrained Gaussian splatting, integrate 3D-2D geometry priors to improve structural accuracy. However, existing methods struggle to capture fine-grained geometric details due to due the loss of high-frequency geometric details during feature learning, which results in limited multi-scale representation. To address this, we introduce a novel wavelet-conditioned implicit SDF model that enhances geometric precision by leveraging a pretrained wavelet autoencoder optimized with sharp depth maps. This autoencoder extracts multi-scale wavelet transformed features, which are fused with implicit 3D triplane features via triplane projection, producing a more structured and detail-preserving distance field. Our method can serve as a plug-and-play module, seamlessly integrating with any implicit SDF representations.

Extensive evaluations on DTU, Tanks and Temples, and a cultural heritage dataset demonstrate that our model consistently outperforms state-of-the-art implicit and explicit 3D reconstruction methods, achieving more complete surfaces with fine-detail preservation across diverse scene scales, from small objects to large architectural buildings.

## 1 Introduction

Image-based 3D reconstruction methods, such as Structure from Motion (SfM), recover 3D structures from multi-view 2D images (Schönberger (2016)), yet they are struggling to preserve high-fidelity details. Alternatively, reconstruction from structured light scans (Hu et al. (2022); Wang et al. (2022)) or a fusion of images and LiDAR scans (Moemen et al. (2020)) has seen active progress, but these point-based methods are prone to noise in scans, making it difficult to obtain plausible reconstruction mesh. High-fidelity reconstruction with fine structural details remains a core challenge in computer vision. Recent advances in implicit representations, such as neural radiance fields (NeRF) by Mildenhall et al. (2021), and explicit methods such as Gaussian splatting (GS) proposed by Kerbl et al. (2023), have significantly advanced 3D applications.

Implicit models leverage photometric consistency loss to learn Signed Distance Fields (SDFs) from multi-view images (Hasson et al. (2020)). Unisurf by Oechsle et al. (2021) unifies surface and volume rendering to improve generalization, while hybrid volume-surface representations can be converted into high-quality meshes for real-time rendering , like the work by Yariv et al. (2023). Multi-resolution hash grids further enable coarse-to-fine optimization for detailed neural surface reconstruction (Li et al. (2023b)), making implicit SDF models effective for complex topologies and continuous geometry fields.

Kerbl et al. (2023) use Explicit Gaussian splatting (GS) to represent scenes using anisotropic 3D Gaussians, enabling efficient training and real-time rendering. However, while GS offers speed, it often sacrifices geometric quality. To address this, AGS-Mesh by Ren et al. (2025) incorporates meshing priors, PGSR proposed by Chen et al. (2024) enforces planar constraints, Turkulainen et al. (2024) utilize depth and normal priors for DN-Splatter method, and 2D GS by Huang et al. (2024a) simplifies 3D Gaussian parameters to improve surface alignment.

Most prior work emphasizes global shape reconstruction and coarse geometric structures. While some methods incorporate geometric priors to enhance shape representation, they often struggle with fine-grained details due to high-frequency feature loss, as current network architectures have limited band representation capacity, often requiring complex 3D prior integration. To overcome these challenges, we propose a multi-scale wavelet-based feature approach utilizing a pre-trained depth image autoencoder trained on monocular depth priors. Wavelets efficiently capture high-frequency geometric details while preserving spatial localization, unlike Fourier transforms, which lose spatial information. This property is crucial for retaining fine surface details that deep learning models often neglect due to the lack of specialized multi-scale representation. The autoencoder is trained on wavelet-transformed depth images generated by a state-of-the-art monocular depth diffusion model (He et al. (2024)). The extracted wavelet features are aligned with implicit 3D triplane features via triplane projection and fused to enhance SDF predictions. Our method outperforms state-of-the-art reconstruction models across diverse scenes. The main contributions of our work can be summarized as follows:

- **Wavelet-Transformed Depth Feature Conditioning**: We introduce a pre-trained multi-scale wavelet autoencoder for depth image reconstruction. During implicit SDF training, wavelet features extracted from depth maps condition the network, enhancing geometric detail preservation.

- **Triplane-Aligned Wavelet Feature Projection**: A triplane projection strategy aligns 2D wavelet features with 3D implicit representations, ensuring seamless fusion and improved geometric consistency.

- **Hybrid Feature Fusion for SDF Prediction**: A UNet-based fusion mechanism integrates implicit 3D features with wavelet-transformed depth representations, yielding more structured and accurate SDF predictions for isosurface mesh extraction.

## 2 Related Work

**Geometry Representation.** 3D geometry representation follows two main paradigms: implicit and explicit. Implicit methods model surfaces via neural radiance fields (NeRF) (Mildenhall et al. (2021)), surface reconstruction (Unisurf by Oechsle et al. (2021)), or signed distance functions (BakedSDF by Yariv et al. (2023)). Explicit methods, such as Structure from Motion (SfM by Schönberger (2016)) and Multi-View Stereo (MVS by Shen (2013)), reconstruct 3D geometry from multi-view images. Recent advances, like 3D Gaussian Splatting (Kerbl et al. (2023)), enable real-time rendering while maintaining high fidelity. Each approach balances reconstruction accuracy, efficiency, and rendering quality.

Further refinements address aliasing artifacts, such as Mip-NeRF created by Barron et al. (2021), and Mip-NeRF 360 created by Barron et al. (2022), which extends NeRF-based models to large-scale unconstrained environments (NeRF in the Wild proposed by Meshry et al. (2019)). Implicit SDF models reconstruct shapes from single images (DISN by Xu et al. (2019)) and enhance local geometry with SDF priors by Chabra et al. (2020).

Recent work integrates SDFs with diffusion models for high-fidelity shape generation from text or single image input (Shim et al. (2023); Zheng et al. (2023); Chou et al. (2023); Li et al. (2023a); Cheng et al. (2023)). Explicit Gaussian-based methods (Kerbl et al. (2023)) continue evolving: AGS-Mesh (Ren et al. (2025)) incorporates meshing priors, PGSR (Chen et al. (2024)) enforces planar constraints for structured Gaussian point clouds, and DN-Splatter (Turkulainen et al. (2024)) integrates depth and normal supervision for improved reconstruction.

**Spectrum Techniques.** Spectral methods have long played a crucial role in computer vision. The frequency analysis of a Fourier Transform has inspired spectral convolution kernels in CNNs like the work by Lavin & Gray (2016) and enabled Fourier Convolutional Neural Networks (FCNN) by Pratt et al. (2017). Similarly, the Fourier transform has also been integrated into multi-head attention mechanisms for Fourier Transformers (He et al. (2023); Nguyen et al. (2022); Buchholz & Jug (2022)), enhancing image feature learning. However, Fourier-based features often face training challenges due to the broad frequency distribution and the loss of locality.

Wavelet transforms provide a more localized spectral representation, preserving spatial details lost in the standard Fourier transform. They have been widely applied in image denoising (Mohideen et al. (2008); Chang et al. (2000)), super-resolution (Guo et al. (2017); Huang et al. (2017)), and restoration, as well as compression (Shen & Delp (1999); Rippel & Bourdev (2017); Ma et al. (2020)) and inpainting (Huang et al. (2024c); Yu et al. (2021); Figueiredo & Nowak (2003)). Wavelet autoencoders (Fujieda et al. (2018); Chen et al. (2018); Mishra et al. (2020); Sadat et al. (2024); Schelkens et al. (2003)) efficiently represent image features while reducing parameters for lightweight models.

Recent advances extend spectral methods to 3D tasks. Sitzmann et al. (2020) leverage periodic activation functions in implicit MLPs to capture repeating geometric patterns, while Liu et al. (2024) adapts spectral variables for feature learning. Fourier bases have been explored for implicit representations by Li et al. (2024), with models like Bacon by Lindell et al. (2022) and BANF by Shabanov et al. (2024) make use of progressive learning for band-limited feature capture in 3D reconstruction.

Wavelets have also been integrated into multi-scale triplane radiance fields (Khatib & Giryes (2024)) and SDF diffusion models (Hu et al. (2024); Zhou et al. (2024); Hui et al. (2022)), enhancing shape generation with fine-grained local details. Despite these advances, spectral models still face convergence challenges, and implementing wavelet decomposition in 3D feature spaces remains computationally demanding.

## 3 Method

The proposed reconstruction method leverages implicit triplane features $\mathbf{F}_{xy}, \mathbf{F}_{xz}, \mathbf{F}_{yz}$, whihc are learned 2D feature grids aligned with three orthogonal planes to encode both geometric and appearance information of a 3D scene. As shown in Figure 1, our framework utilizes these triplane representations for efficient 3D reconstruction from images with known pose. For any 3D query point along a sampled ray, features are retrieved from the three orthogonal planes and aggregated to predict the Signed Distance Function (SDF) value at that location.

To enhance this representation, we introduce a pipeline that enriches triplane features with wavelet-encoded geometric details extracted from input images $\mathbf{X}_i$. These subset input images undergo monocular depth estimation and multi-resolution wavelet transforms before being aggregated and fused into refined triplane features $\{\mathbf{F}_{xy}^{fused}, \mathbf{F}_{xz}^{fused}, \mathbf{F}_{yz}^{fused}\}$ for high-quality SDF prediction. In essence, our method improves surface reconstruction by integrating implicit triplane features with multi-scale wavelet features. The following sections detail each stage of the method along with its mathematical formulation.

### 3.1 Preliminaries

**Implicit Neural Rendering.** Implicit NeRF encodes a 3D scene by representing its volume density and color field, leveraging multi-view posed images through volume rendering. A pixel ray $\mathbf{r}(t) = \mathbf{o} + t\mathbf{d}$ is defined, starting from the camera position $\mathbf{o} \in \mathbb{R}^3$ and traversing along the view direction $\mathbf{d} \in \mathbb{R}^3$. Radiance integration along the ray accumulates color contributions from sampling points of each ray to generate the final pixel color. For each sampling point, volume density $\sigma$ and radiance $\mathbf{c}$ are predicted using separate MLPs. The rendered pixel color $\hat{\mathbf{C}}$ is calculated by $\mathbf{T}(t) = \exp(-\int_{t_n}^{t} \sigma(\mathbf{r}(u))\,du)$, density $\sigma(t)$, and color $\mathbf{c}(t)$ over the ray, bounded by $t_n$ (near) and $t_f$ (far):

$$\hat{\mathbf{C}} = \int_{t_n}^{t_f} \mathbf{T}(t)\sigma(\mathbf{r}(t))\mathbf{c}(t)\,dt. \tag{1}$$

For practical computation, the numerical quadrature-based integration in Alpert (1999) is used to approximate continuous integral calculation.

**SDF-Based Neural Implicit Surface.** A 3D surface $\mathcal{S}$ can be implicitly represented using the zero-level-set of its signed distance function $f(\mathbf{x}) : \mathbb{R}^3 \to \mathbb{R}$, with a 3D point initialized from the depth map of color image $\mathbf{X}$ as input. Here, $\mathcal{S} = \{\mathbf{x} \in \mathbb{R}^3 \,|\, f(\mathbf{x}) = 0\}$, can be seen as the zero-crossing of the signed distance

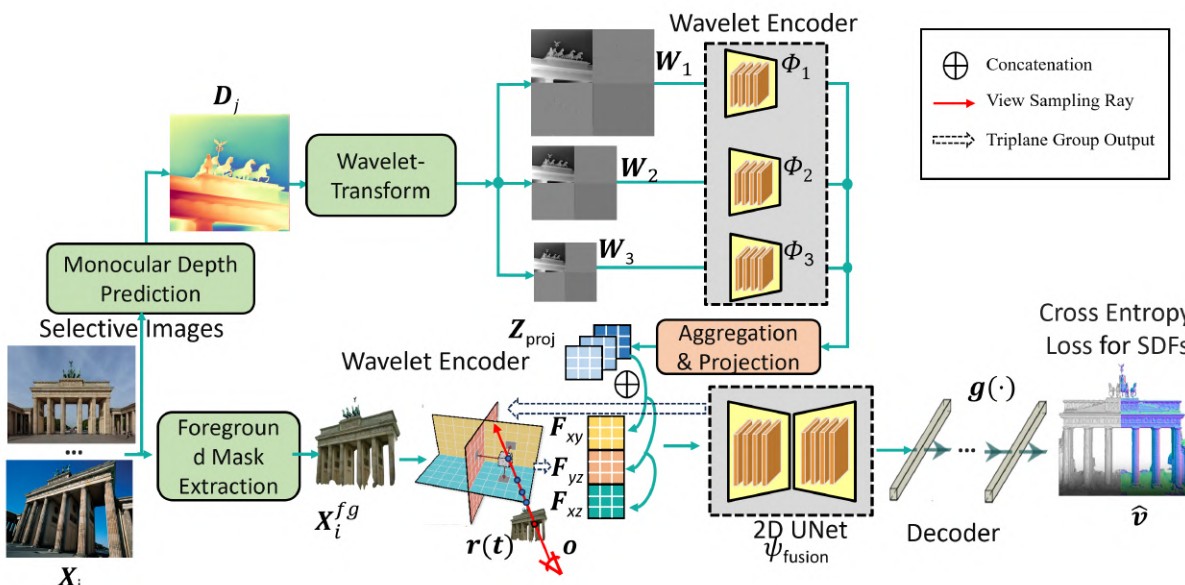

Figure 1: Our model is based on implicit triplane feature fusion for Signed Distance Function (SDF) prediction. Given an input view image $\mathbf{X}_i$, a foreground mask $\mathbf{X}_i^{fg}$ is extracted to focus on the target region for SDF queries. For each pixel, its ray is traced from the camera view $\mathbf{C}_i$ to query the implicit triplane features $\{\mathbf{F}_{xy}, \mathbf{F}_{xz}, \mathbf{F}_{yz}\}$. Images with close-up details are processed via a monocular depth prior to predict depth maps, followed by wavelet transforms in three resolutions. The transformed features $\mathbf{W}_*$ are encoded through a multi-scale wavelet feature encoder $(\Phi_1, \Phi_2, \Phi_3)$ and aggregated into a fused wavelet feature map. This map is projected onto three orthogonal planes, producing $\mathbf{Z}_{proj} = \{\mathbf{Z}_{xy}, \mathbf{Z}_{xz}, \mathbf{Z}_{yz}\}$. The triplane features (Peng et al. (2020)) and wavelet features are concatenated and further fused using a 2D U-Net $\psi$. Finally, MLPs $\mathbf{g}(\cdot)$ decode the fused features to predict SDF values. During inference, the isosurface is extracted via marching cubes to generate the mesh.

function. NeuS by Wang et al. (2021) reformulates volume density rendering in NeRF into a signed distance field (SDF) representation by employing a logistic function to optimize for neural volume rendering,

$$\sigma(\mathbf{x}) = \phi_s(f(\mathbf{x})), \tag{2}$$

where $\phi_s(x) = se^{-sx}/(1 + e^{-sx})^2$ is a logistic density function. It can be derived as the derivative of the sigmoid function $\Phi_s(x) = (1 + e^{-sx})^{-1}$, and is parameterized by the slope $s$. The final opaque density $\sigma(t)$ along the ray is thus given by:

$$\sigma(t) = \max\left( -\frac{d\Phi_s}{dt}(f(\mathbf{r}(t)))/\Phi_s(f(\mathbf{r}(t))), 0 \right). \tag{3}$$

## 3.2 Preprocessing

To get rid of clutter pixels like humans and animals existing in the random online images of landmarks collected in the wild, we further utilize a preprocessing pipeline to create cleaner image and masks for training high-fidelity reconstruction meshes.

The preprocessing pipeline including distractor detection, distractor mask, background mask, effectively filters out non-architectural elements to focus the training only on the relevant structural components. The end result provides clean input data where query rays are only generated for the actual building geometry, improving the quality of the learned implicit representation.

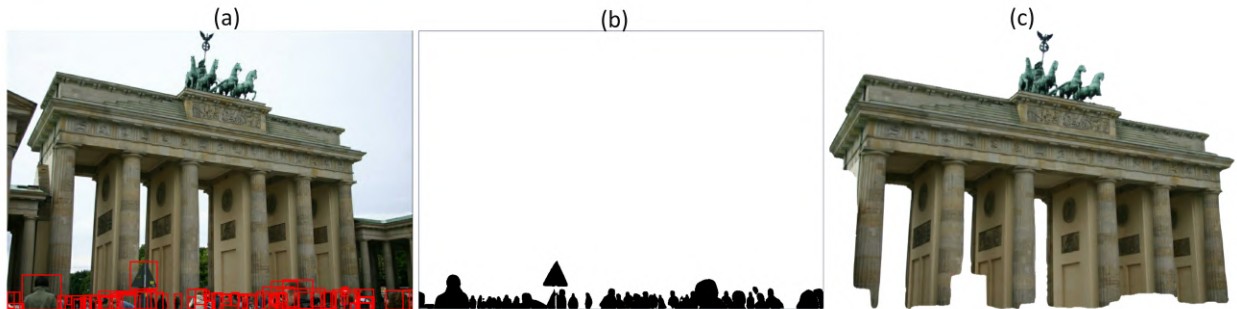

Figure 2: A three-stage preprocessing pipeline for distractor removal: (a) Initial detection identifies unwanted elements like people and objects in the foreground using object detection, (b) Segment Anything Model (SAM) created by Kirillov et al. (2023) converts these detections into precise segmentation masks shown in black silhouettes, and (c) The final masked result isolates the architectural structure by removing both the detected distractors and sky background, leaving only the foreground building pixels that will be used for training the implicit model given query rays.

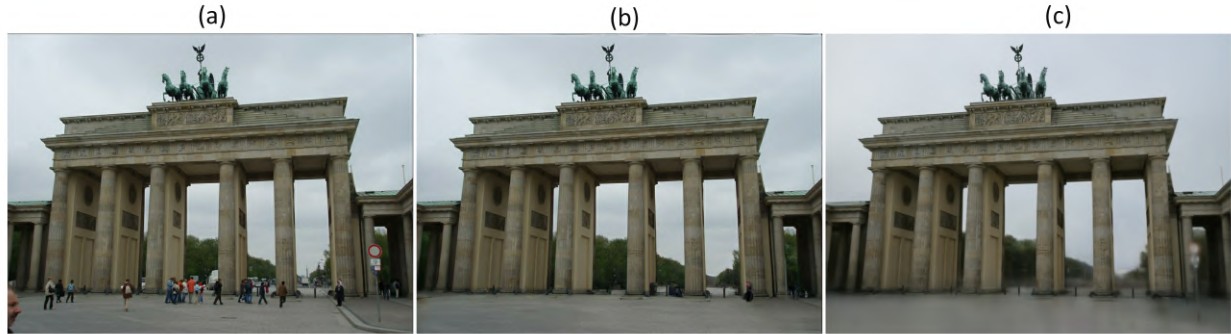

Figure 3: (a) Raw image with distractor on the ground. (b) Inpainted image without distractor as training input. (c) Rendered color image predicted by conditioning on trained implicit SDF model. The whole distractor removal process on the raw image is followed by diffusion model. In the end, our implicit model after training can render the full image without the distractor pixels.

Furthermore, our model can also be directly used to render the clean color image by introducing a color rendering head. The images in Figure 3 show a comparison of removing unwanted pixels (like people) from a photo of the Brandenburg Gate in Berlin. The input to our framework is a color image (a), which contains pedestrians in front of the Berlin Gate. This image serves as the initial scene for further cleanup. In the next step, distractors are detected and removed, followed by an inpainting process to recover the missing pixels, resulting in the processed image (b). Finally, (c) presents the rendered output generated by the pre-trained implicit model, demonstrating the scene reconstruction without distractors and validating the effectiveness of our approach. Such color rendering result is implemented by introducing an additional head for color prediction conditioned on the SDF value prediction of the original implicit 3D model to justify the clean 3D representation of implicit SDF model. This paper is still focused on the results of the 3D reconstruction instead of the results of the rendering.

Wavelet transforms are applied selectively to high-quality close-up images to optimize training efficiency for wavelet feature fusion.

The overall pipeline effectively removes the transient elements (people) while preserving and reconstructing the underlying static architecture through a combination of detection, masking and inpainting for a cleaner 3D reconstruction.

### 3.3 Model Structure

We adopt the same implicit volumetric rendering expression as clarified in previous section for the following model introduction. The whole model is composed of five parts, including a multi-scale wavelet feature encoder, a triplane feature query, a wavelet encoder with output feature projection onto triplane, a triplane feature fusion, and an implicit SDF decoder. In particular, the input of wavelet encoder is monocular depth, while all multi-view color images are used as input to the triplane feature encoder.

**Wavelet Encoder for Multi-Scale Features.** Given a selected input image $\mathbf{X}_i \in \mathbb{R}^{H \times W \times 3}$ from the original multiview images, the monocular depth prior of selected images predicts a depth map $\mathbf{D}_i \in \mathbb{R}^{H \times W}$. The selection of particular close-up images is based on the quality and details that exist in the input view image, and most views are quite distant with blurry pixels, thus making it hard for the monocular depth estimation to provide accurate depth details. The depth map $\mathbf{D}_i$ undergoes a wavelet transform in three resolutions to produce multi-scale wavelet features $\{\mathbf{W}_1, \mathbf{W}_2, \mathbf{W}_3\}$. These are generated using a wavelet transform in three resolutions. These features are then processed by the wavelet encoder $\Phi$ with three different sizes, resulting in a final fused wavelet feature map $\mathbf{Z}_{wave} \in \mathbb{R}^{H' \times W' \times C}$ through feature aggregation:

$$\mathbf{Z}_{wave} = \Phi_1(\mathbf{W}_1) + \Phi_2(\mathbf{W}_2) + \Phi_3(\mathbf{W}_3), \tag{4}$$

where $\Phi_{1,2,3}$ are scale-specific encoding functions to extract various sized features. Usually, $C$ is four channels, representing 2D signals through four filters, defined as **LL**, **LH**, **HL**, and **HH**. Given an input image $\mathbf{X}$, the 2D wavelet transform with specific scale decomposes the image into a low-frequency component $\mathbf{x_L}$ and three high-frequency components $\{\mathbf{x_H}, \mathbf{x_V}, \mathbf{x_D}\}$, corresponding to horizontal, vertical, and diagonal details respectively.

We train the wavelet feature encoder using an autoencoder similar to the design of LiteVAE (Sadat et al. (2024)), aiming to reconstruct the original depth map from its wavelet-transformed representation. We apply a single-level wavelet decomposition independently at multiple scales of the input depth map, generating multi-scale wavelet feature maps. This allows the encoder to capture fine-to-coarse spatial details efficiently. After pretraining three separate wavelet encoders, we obtain their extracted multi-scale feature representations. To ensure alignment, We downsample the feature maps from the two higher-resolution encoders by factors of $1/2$ and $1/4$, respectively, to align with the smallest-scale feature map. This downsampling and aggregation follow the same process as Sadat et al. (2024) proposed in LiteVAE. To mitigate the loss of fine details, we retain multi-scale information by aggregating features across different resolutions. Furthermore, since each wavelet decomposition produces four sub-bands (LL, LH, HL, HH), we stack these submaps along the channel dimension before passing them to the subsequent processing pipeline.

We provide example results of wavelet transformed features in Figure 4, which demonstrates the effectiveness of wavelet transforms in preserving geometric information from depth maps. The visualization compares original depth maps $\mathbf{D}_j$ (top row) with their corresponding wavelet decompositions $\mathbf{W}_j$ (bottom row). The input depth maps, predicted by the state-of-the-art diffusion-based monocular depth estimation network of LOTUS by He et al. (2024), capture detailed geometric structures and continuous depth variations. Our wavelet transform decomposes these depth maps in three resolutions into multi-scale feature representations $\mathbf{W}_j, j = 1, 2, 3$ with three levels, where each level $j$ preserves both spatial and frequency information critical for geometric detail reconstruction. This multi-resolution representation enables the model to effectively encode both fine-grained surface details and global shape features. The wavelet transform decomposes each depth map with a specific resolution into four sub-bands (LL, LH, HL, HH), effectively capturing different frequency components. While LL retains global structure, LH and HL emphasize horizontal and vertical details, and HH captures diagonal features. This highlights the ability to preserve and distinguish depth-specific geometry, and such spatial can also be easily aligned with the image feature map.

**Pixel Ray Query for Implicit Triplane Features.** Each pixel of the foreground masked input image $\mathbf{X}_i^{fg}$ is associated with a ray cast from the camera view $\mathbf{o} \in \mathbb{SE}(3)$. All the sampled points along query rays of each image retrieves implicit triplane features $\{\mathbf{F}_{xy}, \mathbf{F}_{xz}, \mathbf{F}_{yz}\}$ from three orthogonal planes $\{xy, xz, yz\}$ of the 3D space via ray projection, where each plane has a feature resolution of $\mathbb{R}^{H' \times W' \times C'}$, with feature channel dimension $C'$:

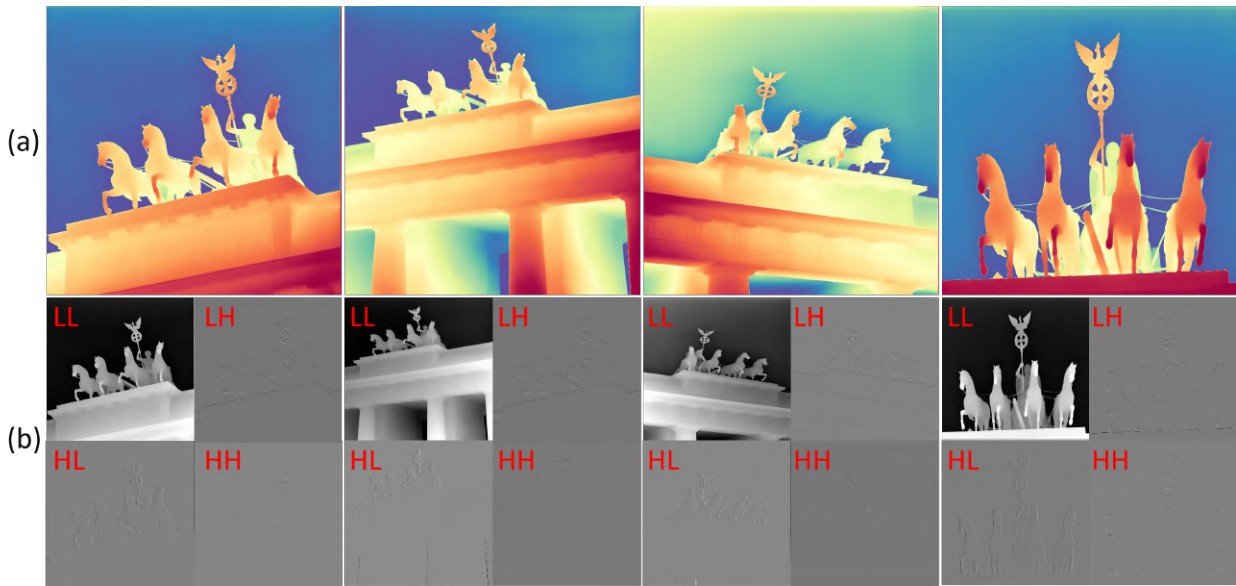

Figure 4: Wavelet transform of depth map in finest resolution, (a) is the original depth map, and (b) is the wavelet transform of the depth map, composed of four parts: Low-Low (LL), Low-High (LH), High-Low (HL), and High-High (HH). The wavelet transformed depth is used as input for the AutoVAE Encoder.

$$\{\mathbf{F}_{xy}, \mathbf{F}_{xz}, \mathbf{F}_{yz}\} = \text{Query}_{\{xy,xz,yz\}}(\mathbf{r}(t)). \tag{5}$$

**Wavelet Feature Projection onto Triplane.** Meanwhile, the wavelet feature map $\mathbf{Z}_{wave}$ of Equation 4 is projected onto the three orthogonal 2D planes to match the implicit triplane feature resolution. This cosine projection generates three projected wavelet feature maps $\{\mathbf{Z}_{xy}, \mathbf{Z}_{xz}, \mathbf{Z}_{yz}\}$ respectively.

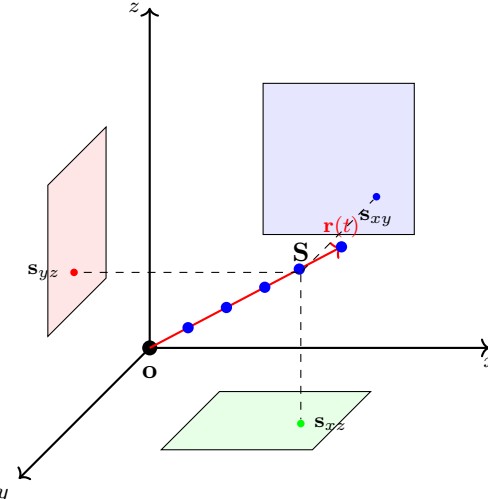

Figure 5: Sampling points along the pixel ray $\mathbf{r}(t)$ starting from $\mathbf{o}$ for implicit triplane feature learning via projection. For Wavelet feature projection onto triplane. The ray $\mathbf{r}(t)$ starts from the camera origin $\mathbf{o}$, then passes through a single unprojected point $\mathbf{S}$. Dashed lines represent the orthogonal projections onto the $xy$, $xz$, and $yz$ planes to obtain triplane features $s_{xy}$, $s_{xz}$, and $s_{yz}$ for a 3D point.

To incorporate wavelet features into the implicit Signed Distance Field (SDF) model, we first generate a structured 3D representation of the scene by leveraging aligned depth maps. Specifically, we reconstruct a dense unprojected point cloud in the camera frame followed by camera to world transform. This transformation involves back-projecting depth pixels into 3D space using the known intrinsic and extrinsic camera parameters. The resulting 3D points are then associated with wavelet-based features aligned with 2D pixels.

Once the wavelet-enhanced feature map is obtained, it is projected onto the three feature-aligned triplane representations corresponding to the orthogonal planes defined by the normal vectors $(1, 0, 0), (0, 1, 0), (0, 0, 1)$. This projection ensures that the 3D wavelet features are properly integrated into the implicit triplane feature space. Mathematically, this process is formulated as follows:

$$\{\mathbf{Z}_{xy}, \mathbf{Z}_{xz}, \mathbf{Z}_{yz}\} = \mathbf{P}\mathbf{Z}_{wave} \cdot \cos(\{\alpha, \beta, \gamma\}), \tag{6}$$

where $\{\mathbf{Z}_{xy}, \mathbf{Z}_{xz}, \mathbf{Z}_{yz}\}$ represent the projected wavelet-enhanced features on the three orthogonal feature planes $xy, xz, yz$. The projection angles $\alpha, \beta, \gamma$ correspond to each feature plane, ensuring an optimal alignment between the wavelet features and the triplane encoding. Such angle is the dot product between ray direction and axis direction. The transformation matrix $\mathbf{P}$, derived from the camera intrinsic parameters and the camera-to-world extrinsic pose, maps unprojected 3D points from the camera frame to the world coordinate system. These mapped points are then projected onto the triplane feature planes, where they serve as inputs to our method. Each pixel ray maps 3D spatial information onto a set of triplane feature representations. Given a camera pixel ray $\mathbf{r}(t)$, we analyze the sampled point $\mathbf{S}$ along the ray scaled by the predicted depth value and compute its orthogonal projections onto the three principal planes: $xy$, $xz$, and $yz$. These projections provide the corresponding triplane feature locations $\mathbf{s}_{xy}$, $\mathbf{s}_{xz}$, and $\mathbf{s}_{yz}$.

Figure 5 illustrates the feature extraction process along a pixel ray $\mathbf{r}(t)$. The ray originates from the camera at $\mathbf{o}$, extends through the sampled point $\mathbf{S}$, and continues along its trajectory. Dashed lines indicate the orthogonal projections of $\mathbf{S}$ onto the three triplane feature planes ($xy$, $xz$, and $yz$), which are used for feature representation.

In the Figure 5, implicit triplane features are obtained by sampling multiple points along the pixel ray uniformly, capturing continuous spatial information. Wavelet triplane features are projected in the same way as the implicit features. A key distinction is that each feature plane in the implicit approach consists of 16 channels, whereas the wavelet-based plane features are compressed into 4 channels, reducing redundancy while preserving essential spatial details.

This structured feature projection enables a seamless integration of 2D wavelet-transformed depth features with 3D implicit features, leading to more accurate SDF predictions and higher-fidelity 3D reconstructions.

**Feature Concatenation and Fusion.** The implicit triplane features and the projected wavelet features of each feature plane are concatenated along the channel dimension and fused using a 2D U-Net. This results in three fused triplane features $\{\mathbf{F}_{xy}^{fused}, \mathbf{F}_{xz}^{fused}, \mathbf{F}_{yz}^{fused}\}$, where $\mathbf{F}_{*}^{fused} \in \mathbb{R}^{H' \times W' \times 2C}$:

$$\{\mathbf{F}_{xy}^{fused}, \mathbf{F}_{xz}^{fused}, \mathbf{F}_{yz}^{fused}\} = \{\psi_{\text{fusion}}([\mathbf{F}_{xy}; \mathbf{Z}_{xy}]),$$
$$\psi_{\text{fusion}}([\mathbf{F}_{xz}; \mathbf{Z}_{xz}]), \psi_{\text{fusion}}([\mathbf{F}_{yz}; \mathbf{Z}_{yz}])\}, \tag{7}$$

where $[;]$ denotes concatenation of feature maps along the channel dimension.

**Encoder for SDF Prediction.** The fused features are finally decoded by a neural network $g(\cdot)$, which predicts the SDF value $v \in \mathbb{R}$ for the given pixel ray query:

$$v = g(\mathbf{F}_{xy}^{fused}, \mathbf{F}_{xz}^{fused}, \mathbf{F}_{yz}^{fused}). \tag{8}$$

The predicted SDF values are used to extract the isosurface via marching cubes created by Lorensen & Cline (1998), producing a reconstructed 3D mesh.

**Loss Function.** The total training loss $\mathcal{L}_{\text{total}}$ for the implicit model is defined as a combination of three components: the mean cross-entropy loss $\mathcal{L}_{\text{CE}}$, the Eikonal regularizer $\mathcal{L}_{\text{Eik}}$, and the depth loss $\mathcal{L}_{\text{depth}}$. The total loss is defined as:

$$\mathcal{L}_{\text{total}} = \mathcal{L}_{\text{CE}} + \lambda_{\text{Eik}}\mathcal{L}_{\text{Eik}} + \lambda_{\text{depth}}\mathcal{L}_{\text{depth}}, \tag{9}$$

where each loss term is defined as follows:

$$\mathcal{L}_{\text{CE}} = -\frac{1}{N}\sum_{i=1}^{N}\Big[v_i\log(\hat{v}_i) + (1 - v_i)\log(1 - \hat{v}_i)\Big], \tag{10}$$

which represents the mean cross-entropy loss computed over $N$ training samples, where $v_i$ is the ground truth signed distance function (SDF) value, and $\hat{v}_i$ is the predicted SDF value.

$$\mathcal{L}_{\text{Eik}} = \frac{1}{M}\sum_{i=1}^{M}\big|\|\nabla\hat{v}_i\| - 1\big|^2, \tag{11}$$

where the Eikon loss is applied to $M$ neighboring sampled points to regularize the smoothness of SDF prediction.

$$\mathcal{L}_{\text{depth}} = \frac{1}{N}\sum_{i=1}^{N}\|\mathbf{D}_i - \hat{\mathbf{D}}_i\|^2, \tag{12}$$

which measures the mean squared error between the predicted depth values $\hat{\mathbf{D}}_i$ and the ground truth depth values $\mathbf{D}_i$. Here, $\lambda_{\text{Eik}}$ and $\lambda_{\text{depth}}$ are weighting factors that balance the contributions of the Eikonal and depth losses, respectively.

In the formulation 9, $\mathcal{L}_{\text{Eik}}$ regularizes the gradients to enforce the local smoothness of signed distance field, and $\mathcal{L}_{\text{depth}}$ ensures consistency with depth observations for better geometry representation learning. For color image rendering, we just need to add another cross-entropy loss and structural similarity loss by Wang et al. (2004) to the color image pixels.

## 4 Experimental Results

**Datasets.** To evaluate the general performance of our 3D reconstruction approach, we make use of a wide variety of datasets, including the DTU (Jensen et al. (2014)) dataset, which is collected from a turntable; the Tanks and Temples dataset (Knapitsch et al. (2017)), captured as video scans of sculptures and buildings; and the Cultural Heritage dataset (Martin-Brualla et al. (2021)), which features large-scale historical sites.

For DTU (Jensen et al. (2014)), we use the Chamfer distance metric calculated between the reconstructed model and the ground truth, while for Tanks and Temples (Knapitsch et al. (2017)), we evaluate reconstruction accuracy using the F1 score ($F_1 = 2 \cdot \frac{\text{Precision} \cdot \text{Recall}}{\text{Precision} + \text{Recall}}$), as Chamfer distance makes it difficult to differentiate the performance for some scenes. Due to the large scene scale of the Cultural Heritage dataset, obtaining ground truth (GT) meshes or pseudo-GT is challenging, so we primarily provide qualitative results. For DTU and Tanks and Temples, we sample 1,000 and 10,000 points uniformly, respectively, and compare them with the nearest 3D points from the GT mesh.

**Baseline models.** For baseline evaluation, we compare our method against several state-of-the-art implicit and explicit 3D reconstruction models. The implicit SDF baselines include VolSDF by Yariv et al. (2021), NeuS by Wang et al. (2021), Neuralangelo by Li et al. (2023b), and BakedSDF by Yariv et al. (2023). The explicit reconstruction baselines include SuGaR by Guédon & Lepetit (2024), GOF by Huang et al. (2024b), and 2DGS by Huang et al. (2024a). We provide quantitative comparisons across DTU and Tanks

and Temples datasets, while qualitative visual comparisons highlight the top three performing models. The input for all baselines are images and camera poses.

**Implementation details.** For monocular depth estimation, we use the diffusion-based LOTUS model by He et al. (2024) to predict depth maps for selected heritage dataset views, while for Tanks and Temples and DTU datasets, we process all training images. Wavelet decomposition is performed using the Fast Wavelet Transform (FWT) (Mallat (1989)) with Haar basis filters.

The autoencoder for wavelet-transformed depth features consists of a ResNet encoder followed by a fully convolutional decoder, similar to LiteAutoVAE (Sadat et al. (2024)). We apply a Gaussian blurring loss to low-frequency sub-bands and a Charbonnier loss (Barron (2019)) to high-frequency sub-bands. During implicit SDF training, the AutoVAE encoder remains frozen. The triplane feature representation is structured as $3 \times 64 \times 64 \times 16$, with an SDF decoder composed of fully connected layers. Wavelet-triplane fusion is achieved via a 2D U-Net with four downsampling and upsampling blocks, followed by a $1 \times 1$ convolution along the depth channel. The fused representation consists of three orthogonal triplane feature planes ($64 \times 64 \times 16$ each), combined with a projected wavelet feature map ($64 \times 64 \times 4$), and refined through the 2D U-Net.

The wavelet autoencoder processes four spectral channels—low-frequency, vertical high-frequency, horizontal high-frequency, and diagonal high-frequency—using ResNet blocks. Wavelet transforms are applied to Lotus-generated depth maps at three resolutions, with extracted features used to train the autoencoder. To balance fine-grained details and global features, we incorporate self-modulated convolutional layers (Sadat et al. (2024)). The loss function includes reconstruction, regularization, and adversarial terms (Sadat et al. (2024)). Features are extracted at $256 \times 256$, $128 \times 128$, and $64 \times 64$ resolutions, with higher-resolution features downsampled by $1/4$ and $1/2$ for alignment. For the cultural heritage dataset, we manually selected 100 close-up images to enhance the implicit SDF model with wavelet-transformed features.

For the implicit SDF model, we use the Facto-SDF implementation from SDFStudio by Yu et al. (2022), integrating it with the triplane feature representation as the encoder backbone.

**Training Complexity.** Our training pipeline consists of two stages: training the wavelet encoder and training the implicit SDF conditioned on the frozen wavelet encoder. The wavelet encoder training takes approximately 8 hours on an RTX 3090. For the implicit SDF training of DTU model, training completes in 1-2 hours. As for Tanks and Temples and Cultural Heritage), initial implicit SDF training on color images takes 6-8 hours due to data diversity, followed by 1-2 hours of fine-tuning with wavelet-triplane features.

All experiments were conducted on an NVIDIA RTX 3090 GPU, ensuring efficient training and inference. This modular approach enables scalable learning across datasets of varying sizes and complexities.

### 4.1 Baseline Comparisons

We first provide the qualitative comparison results of the sample targets or scenes in the three datasets, including the qualitative results of the DTU, Tank and Temple and the Cultural Heritage dataset in Figure 6. These datasets are collected via cameras that point towards a target. The selected scans of the Tank and Temple dataset in Figure 7 include mainly the room scan with the camera pointing outward. Furthermore, the quantitative results on the DTU and Tank and Temple datasets are also provided in Table 1 and Table 2, respectively.

As seen in Figure 6, our model can reconstruct fine-grained details on the mesh surface, such as feature details on birds, details of clothes of happy buddha, owl, texts on the Berlin gate. We recommend readers to have a close-up look in red highlighted circles. The 2D Gaussian Splatting seems to struggle to preserve details and also has obvious artifacts and holes on the mesh surface. The 2D GS also fails to reconstruct a mesh of large-scale Berlin gate. Neuralangelo is very good at preserving some details, but still has some artifacts or obstructions as shown on the bottom of the Berlin gate with unexpected blockings, although normals are consistent along with details of texts. BakedSDF has the worst performance, oversmoothing the results, with a smooth surface and loss of details, particularly obvious on the Berlin gate, which even may incur some unexpected reconstruction mesh regions in front of the house.

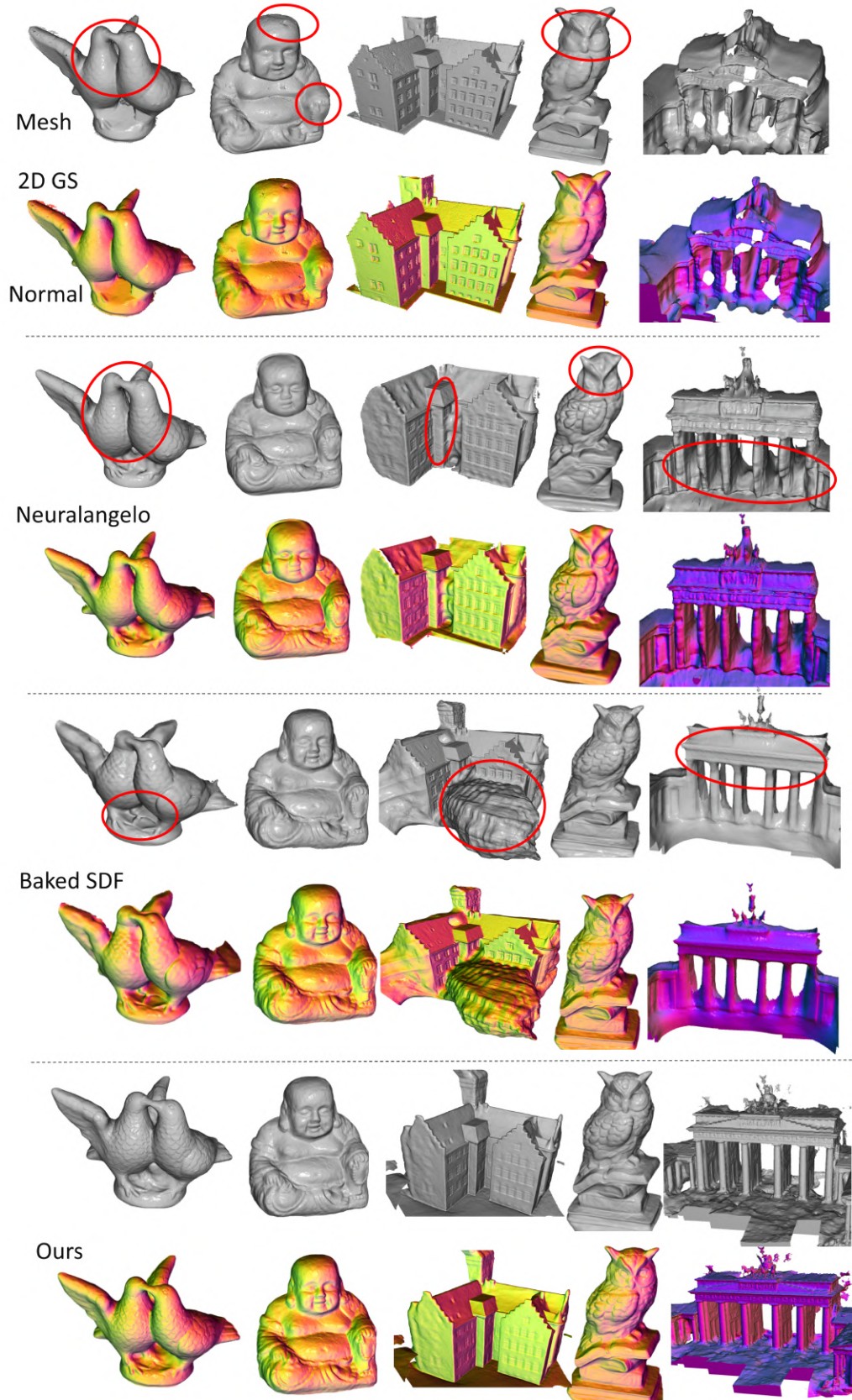

Figure 6: Baseline comparison results on five targets from DTU (Jensen et al. (2014)), and Cultural Heritage (Martin-Brualla et al. (2021)) dataset. Each model result (split by dashed line) contains mesh and normals.

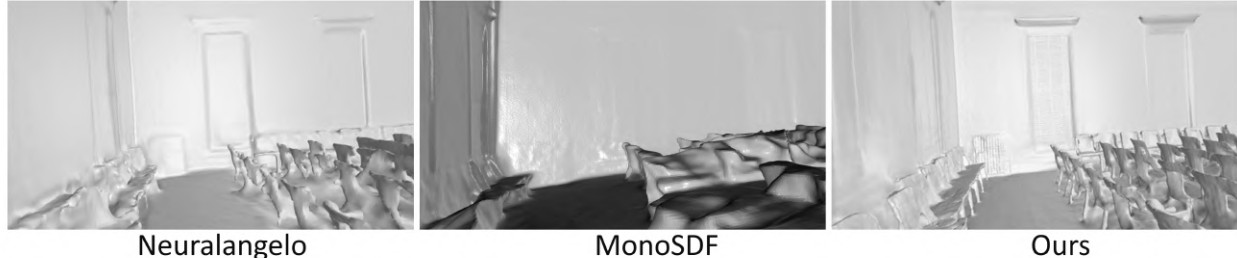

Neuralangelo        MonoSDF        Ours

Figure 7: Baseline comparison results on the inner room scan of Tank and Temple (Knapitsch et al. (2017)). The figures show the zoom-in details of the reconstruction results of the meeting room.

Lastly, the reconstruction of inner room scan with camera pointing outward is presented in Figure 7. While Neuralangelo or MonoSDF either fails to preserve the geometry details, or smooth the details (as exhibited by the middle figure with a flat wall without door), our model preserves the high-fidelity details.

Finally, we provide quantitative evaluation results in Table 1 and Table 2 using chamfer distance and F1 score metric respectively. On DTU, our model is the best, while Neuralangelo gets the second best performance. On Tank and Temple dataset, our model still scores best on three out of four scans while Neuralangelo follows next.

Table 1: Performance evaluation comparison across baselines by chamfer distance ↓ metric. Smaller values indicate better accuracy. Green is the best, while Orange and Yellow indicate second and third best.

| Scan id | 24 | 40 | 55 | 65 | 83 | 97 | 105 | 106 | 110 | 114 |
|---|---|---|---|---|---|---|---|---|---|---|
| VolSDF (Yariv et al. (2021)) | 1.14 | 0.81 | 0.49 | 0.70 | 1.29 | 1.18 | 0.70 | 0.66 | 1.08 | 0.42 |
| NeuS (Wang et al. (2021)) | 1.00 | 0.93 | 0.43 | 0.65 | 1.48 | 1.09 | 0.83 | 0.52 | 1.20 | 0.35 |
| Neuralangelo (Li et al. (2023b)) | 0.37 | 0.35 | 0.35 | 0.54 | 1.29 | 0.97 | 0.73 | 0.47 | 0.74 | 0.32 |
| BakedSDF (Yariv et al. (2023)) | 0.63 | 0.58 | 0.40 | 0.52 | 1.37 | 1.06 | 0.81 | 0.56 | 0.82 | 0.38 |
| SuGaR (Guédon & Lepetit (2024)) | 1.47 | 1.13 | 0.61 | 1.71 | 1.63 | 1.62 | 1.07 | 0.79 | 2.45 | 0.98 |
| GOF (Huang et al. (2024b)) | 0.50 | 0.37 | 0.37 | 0.74 | 1.18 | 1.29 | 0.68 | 0.77 | 0.90 | 0.42 |
| 2DGS (Huang et al. (2024a) ) | 0.48 | 0.39 | 0.39 | 0.83 | 1.36 | 1.27 | 0.76 | 0.70 | 1.40 | 0.40 |
| Ours | 0.45 | 0.34 | 0.32 | 0.54 | 0.97 | 0.82 | 0.54 | 0.55 | 0.68 | 0.27 |

Table 2: Quantitative results of F1 Score↑ for the reconstruction on Tanks and Temples dataset. Our method achieves best reconstruction accuracy for building or point outwards scan in the room. Green is the best, while Orange and Yellow indicate second and third best.

| | Barn | Courthouse | Ballroom | Meetingroom |
|---|---|---|---|---|
| **VolSDF** | 0.19 | 0.20 | 0.12 | 0.21 |
| **NeuS** | 0.29 | 0.17 | 0.16 | 0.24 |
| **Neuralangelo** | 0.70 | 0.28 | 0.36 | 0.32 |
| **SuGaR** | 0.14 | 0.08 | 0.06 | 0.15 |
| **BakedSDF** | 0.63 | 0.23 | 0.19 | 0.13 |
| **GOF** | 0.51 | 0.28 | 0.30 | 0.28 |
| **2D GS** | 0.45 | 0.13 | 0.26 | 0.18 |
| **Ours** | 0.67 | 0.56 | 0.41 | 0.34 |

## 4.2 Ablation Study

The wavelet feature encoder can effectively capture fine-grained details from input views, such as edge features, as shown in Figure 9. The output feature maps of wavelet encoder provides a rich representation of

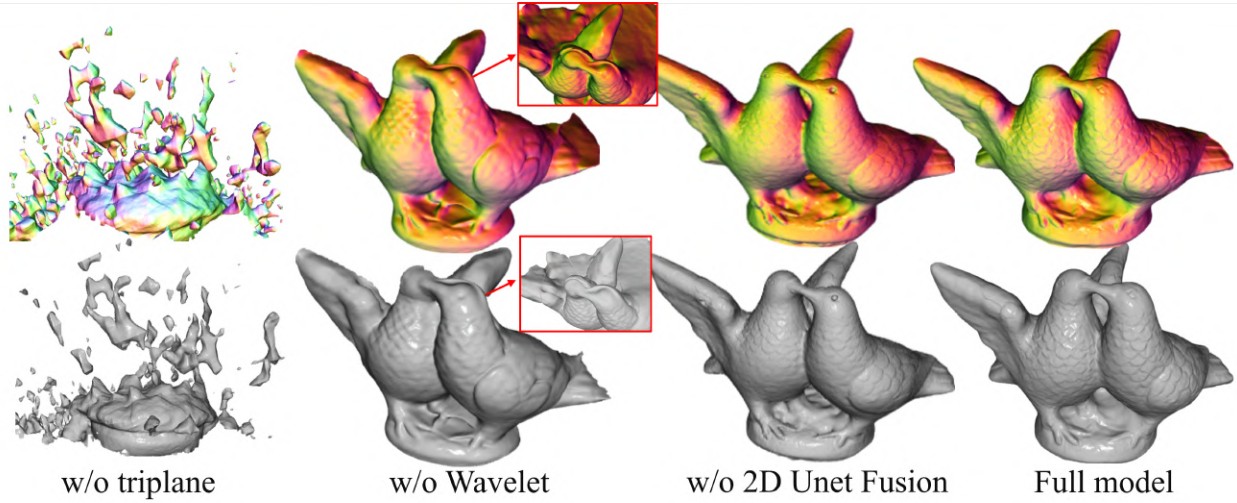

w/o triplane    w/o Wavelet    w/o 2D Unet Fusion    Full model

Figure 8: Ablation study on 3D reconstruction. From left to right: (1) Removing triplane leads to fragmented geometry. (2) Without the wavelet encoder, fine details are distorted. (3) Omitting 2D U-Net fusion results in less sharp features. (4) The full model achieves the best quality.

the input views to encode more geometry details, like the carved letters of front side of the gate. The final reconstruction mesh details can be enhanced by the decoder conditioning on the wavelet features. Figure 9 showcases the feature output of the largest wavelet encoder in an autoencoder pre-trained separately on the wavelet transformed depth input in three resolutions, highlighting the progressive decomposition of image features across multiple frequency bands. The wavelet encoder outputs are visualized across four columns in Figure 9. Column (a) shows the encoded depth features, preserving the overall geometric structure. Column (b) displays vertical gradient features $\Phi$ that highlight edge transitions along the y-axis. Column (c) presents horizontal gradient features, capturing edge variations along the x-axis. Column (d) shows diagonal gradient features that encode diagonal directional geometric variations. This learned decomposition through our wavelet encoder enables comprehensive feature extraction at multiple orientations, crucial for accurate 3D surface reconstruction. Each component contributes specific directional information, allowing the model to capture both directional surface variations and overall geometric structure.

Table 3: Ablation study of our model on DTU (Jensen et al. (2014)), including the main component design and the various loss.

| Metric | w/o Triplane Feature | w/o Multi-scale Wavelet Feature | w/ Single Scale Wavelet | w/o UNet Channelwise Fusion |
|---|---|---|---|---|
| F1-Score↑ | 0.24 | 0.32 | 0.35 | 0.39 |
| Metric | w/o Cross Entropy Loss | w/o Depth Loss | w/o Eikon Loss | Full Model |
| F1-Score↑ | 0.18 | 0.41 | 0.43 | 0.50 |

We conduct ablation studies to assess the contribution of each key component in our model structure, with qualitative results in Figure 8 and quantitative metrics in Table 3.

First, we evaluate the impact of removing the triplane representation, leaving only an MLP-based implicit function for Signed Distance Field (SDF) modeling. This results in severe fragmentation, geometric artifacts, and disconnected surfaces, reflected in a sharp F1-score drop to 0.24, underscoring the triplane role in capturing global spatial structure. Next, removing the wavelet encoder while retaining the triplane representation and SDF decoder preserves overall shape but degrades surface details, introducing irregular bumps, particularly in fine-detail regions (e.g., eyes, beak). The F1-score drops to 0.32. Using a single-scale wavelet improves the F1-score to 0.35, while our full multi-scale wavelet approach further enhances detail preservation, demonstrating the importance of multi-scale feature encoding. Removing the 2D U-Net fusion network

and directly concatenating features results in suboptimal integration, leading to slight blurring and less distinct surface transitions, particularly in high-detail regions. This degrades the F1-score to 0.39, highlighting the necessity of learned feature fusion. Further ablations show that excluding Depth Loss and the Eikon term reduces F1-scores to 0.41 and 0.43, respectively, indicating their contribution to geometric accuracy. Our full model, incorporating triplane encoding, multi-scale wavelet processing, U-Net fusion, Depth Loss, and the Eikon term, achieves the best reconstruction quality, with an F1-score of 0.50. This configuration effectively preserves global structure and fine-grained details while maintaining consistent performance across scales in the DTU dataset.

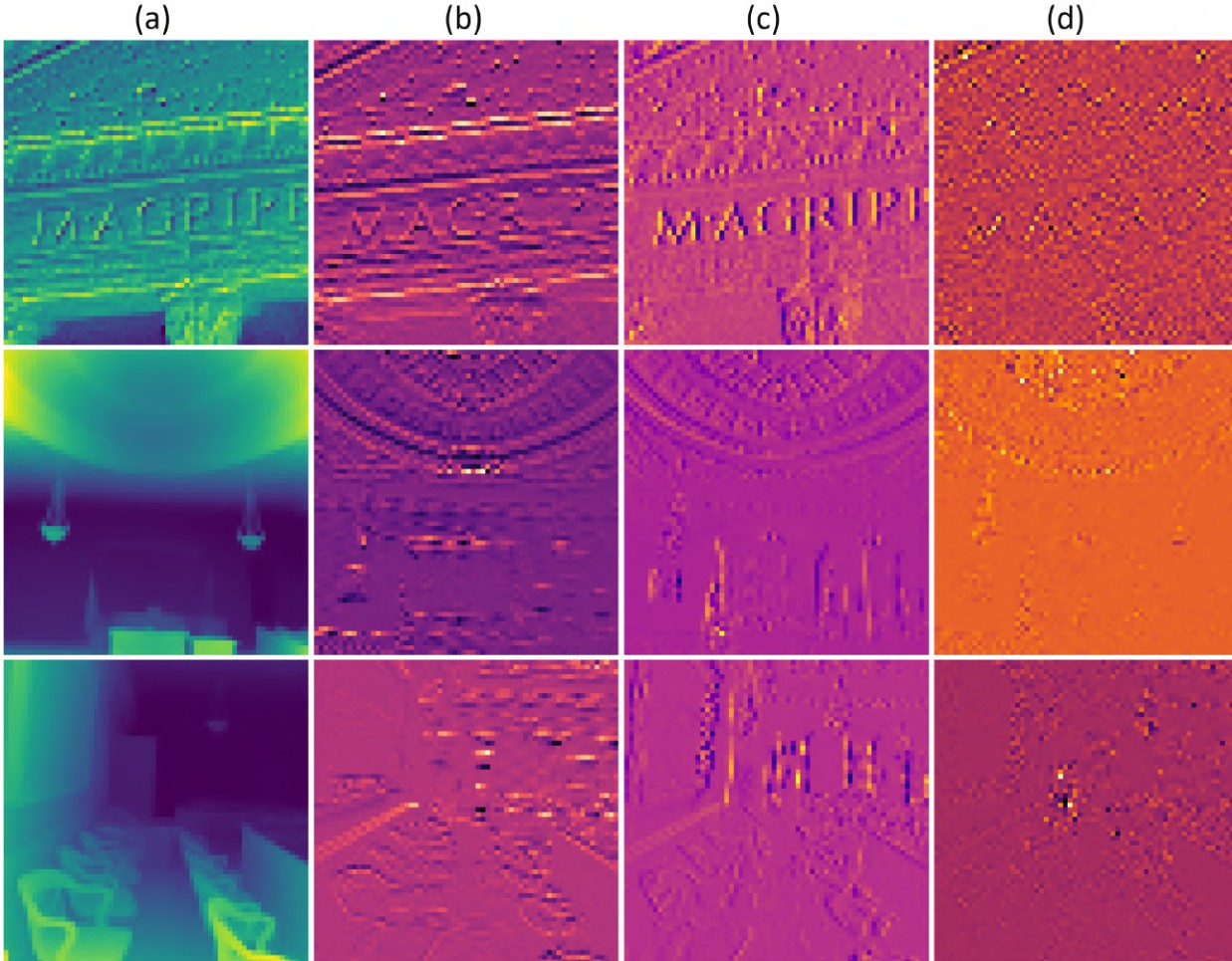

Figure 9: Visualization of learned wavelet encoder feature maps at the highest resolution level. The four columns demonstrate different components of the encoded representation: (a) depth features preserving overall geometric structure, (b) vertical gradient features capturing y-axis surface variations, (c) horizontal gradient features encoding x-axis transitions, and (d) diagonal gradient features representing cross-directional geometric patterns. Each component is processed through our wavelet encoder $\Phi$ to extract orientation-specific geometric information.

## 5 Conclusion

We propose an implicit SDF model that integrates wavelet transformed depth features into a latent triplane feature space. By combining spatially decomposed wavelet representations with triplane embeddings, our approach enhances the preservation of geometric details. During inference, fused features are sampled along query rays and decoded into SDF values, enabling high-fidelity mesh reconstruction. Our model requires only monocular priors from state-of-the-art diffusion-based depth estimation models or a subset of selected

heritage dataset images. Compared to existing implicit SDF and explicit Gaussian Splatting methods, our approach achieves superior shape completeness while retaining intricate geometric details. Despite these advances, opportunities remain for further improvement. Future work could explore optimized sampling strategies to enhance computational efficiency. Additionally, integrating discrete Gaussian representations may accelerate training while maintaining high reconstruction fidelity. These extensions could expand our method applicability to large-scale scenarios and real-time applications.

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
