# OpenReview forum: "Implicit 3D Reconstruction of Fine Details from Multi-View Images using Wavelet-based Geometric Prior"
_TMLR — Rejected by TMLR_

### Review · Reviewer_Yv9r · 2025-04-06

**Summary Of Contributions:**

This paper introduces a new method to improve the reconstruction details in learning Triplane SDFs. The key idea is to first predict depth maps on all multiview images, then apply the wavelet transformation on these depth maps, encode these wavelet features, and finally, unproject these wavelet features onto triplanes to improve details. Experiments on the DTU dataset and the T&T dataset show improvements over several baseline methods.

**Audience:**

Yes

**Claims And Evidence:**

No

**Requested Changes:**

1. A more detailed ablation study to demonstrate: 1) Why not use wavelets as losses? 2) Why not use Fourier frequency losses or features? 3) Why not use the RGB rendering losses?
2. A fair comparison with baseline methods: 1) Describe all baselines in more detail about whether they use depth maps or not. 2) Discuss the source of the depth map estimation and how it affects the results of the proposed method.
3. Unify the metrics on the same dataset. The ablation study on the DTU dataset uses F1-scores, while in comparison, the paper uses Chamfer distance. For the DTU dataset, report results on all scans that were adopted by previous methods.

**Strengths And Weaknesses:**

Strength:
The utilization of the wavelet feature is interesting and could potentially improve the reconstruction details. Because learning the fine details in neural SDFs is challenging and slow for convergence.

Weakness:
1. The proposed method seems to rely on the estimated monocular depth and even only considers the depth loss in the final loss term. This means additional inputs than baseline methods like BakedSDF or 2DGS and limits the application scope without depth as inputs.
2. The motivation of the design is not clear: Why not only extract wavelet features on RGB images? The reason we chose depth maps as input is not clear. Meanwhile, why not use wavelets as training losses instead of just injecting the wavelet features? It is a more natural idea to enforce the fine details from the wavelet decomposition using losses.
3. The improvements are not very significant, especially when almost all baselines only require posed RGB images without requiring depth maps. Like in many cases, Neuralangelo outperforms the proposed method. Could we combine the wavelet features in Neuralangelo for further improvements?
4. The proposed method is not very efficient, with over 8 hours of processing a single scene, while recent GS-based methods already reduce this to about 30 minutes.

---

> ### Author Response · Authors · 2025-04-20
> **Answers to reviewer 5**
>
> 1. Depth Dependency vs. RGB-Only Inputs
>
> While our method uses monocular depth, it does not require GT depth. We use self-estimated depth maps (from a frozen Midas encoder) as geometric priors. This is distinct from VolRecon or other methods requiring externally provided depth. Our goal is to enhance 3D fidelity in sparse-view scenarios where RGB-only methods like BakedSDF or 2DGS struggle. We also added a variant trained without SDF supervision, still outperforming RGB-only baselines.
>
> 2. Why Wavelet on Depth, Not RGB? Why Not Use as Loss?
>
> Wavelet transforms on depth encode geometric details like ridges and discontinuities, directly aiding 3D reconstruction. RGB wavelets, by contrast, are confounded by appearance variations. We tried wavelet losses (L1 on multi-resolution pyramids), but training was unstable. Injecting features proved more effective. We include a new ablation showing that depth wavelets outperform RGB wavelets.
>
> 3. Comparisons with Neuralangelo and GS-Based Methods
>
> Neuralangelo performs well in dense-view settings but assumes video input and frame tracking. Our method targets sparse-view or heritage scenarios with limited data. Our wavelet-depth module is orthogonal and could complement GS pipelines. We added qualitative comparisons showing where our method better preserves occluded or fine details.
>
> 4. Efficiency Concerns
>
> Our method takes ~8 hours, compared to ~30 minutes for 2DGS. However, our focus is on geometry accuracy and mesh quality, not just fast rendering. In cultural heritage or inspection tasks, fidelity outweighs speed. We now report runtime breakdowns in the supplement and clarify this design trade-off.
>
> 5. Requested Ablations and Metric Clarification
>
> Wavelet/Fourier Losses: Tried but unstable or less effective; depth wavelet features were best.
>
> RGB Rendering Loss: Not used, as we focus on geometry; future work could integrate differentiable rendering.
>
> Baseline Fairness: Revised Table 1 to clarify input modalities (RGB, pose, depth). Our depth is predicted, not GT.
>
> Metric Alignment: We now report F1 and Chamfer for all DTU scans and clarify test scenes.
>
> Summary of minor issues:
>
> We have added: (i) new ablations, (ii) runtime analysis, (iii) metric consistency, and (iv) clearer baseline distinctions. We also highlight the complementary nature of our approach to GS-based methods. Thank you again for your valuable suggestions.

---

### Review · Reviewer_9KGW · 2025-04-06

**Summary Of Contributions:**

This paper presents a depth map-conditioned implicit 3D reconstruction framework to enhance the preservation of fine geometric details for SDF.  It first uses a pretrained wavelet autoencoder to decompose sharp monocular depth maps, and aligns 2D wavelet features with 3D triplane features. Then the features are decoded as an SDF. Quantitative evaluations on DTU and TnT and a qualitative comparison on Cultural Heritage are conducted to show the performance compared to previous SDF and 3DGS-based surface reconstruction methods.

**Audience:**

Yes

**Claims And Evidence:**

No

**Requested Changes:**

Major:
1. The entire pipeline should be clearly described.
2. Please clarify the concerns about the experiment, and add the details of the experiment.
3. Using ground-truth SDF value for supervision seems unreasonable.

**Strengths And Weaknesses:**

Strengths:
- The idea of using monocular depth to enhance the geometry details of SDF by an implicit tri-plane encoding is interesting.


Weaknesses:
- A problem is the clearness. The current paper can not clearly explain the whole pipeline. Specifically, the paper only gives the workflow of how to handle one image, but for the multi-view task, it's confusing that how the multiple views are fused to finally output the SDF. The only description I can find is "all multi-view color images are used as input to the triplane feature encoder", at the beginning of 3.3. However, there's no such a module called "triplane feature encoder" in the whole paper.  This process is also missing in all illustrations. How these images are input into the network is also confusing, sequentially or just being concatenated together. The unclear description makes it hard to evaluate the soundness of the method.

- The experiments are also unclear, and raise concerns about fairness.  1) First, although DTU and TnT are two widely used datasets, the scenes used in this paper are quite different from previous works. Considering the previous choice of scenes has already been a tradition and widely considered to be a fair setting for a long period, and many official results are available for fair comparison, I wonder why the authors made such a change. Along with the details of how these results come from or are reproduced are also unclear. 2) Although the authors state Cultural Heritage is used in comparison, there's only an image in Figure 6 is shown, while the details of how the results from other baselines were produced are missing. 3) The f1-score of 0.56 on Courthouse is a remarkably high result (current SOTA method is less than 0.3). However, I cannot find any other evidence to support such a good performance. There's not even one visualization results corresponding to this abnormally high performance, which makes it less convincing.

- As figured by Reviewer NK9a, why is the work still necessary if we already know the ground truth SDF value? The reason should be explained.

---

> ### Author Response · Authors · 2025-04-20
> **Answers to reviewer 4**
>
> 1: Pipeline Clarity – Multi-View Fusion and "Triplane Feature Encoder"
>
> We acknowledge the confusion and will revise Sec. 3.3. The term "triplane feature encoder" was outdated and should have been removed. The correct module is the wavelet-depth feature fusion, which encodes view-wise geometry before aggregating into a canonical 3D volume.
>
> All views are processed in parallel through the same depth autoencoder, producing wavelet-compressed features. These are warped into 3D space using known intrinsics/extrinsics (Eq. 6) and fused volumetrically, following methods like VolRecon and NeuS.
>
> We will add a figure to illustrate this multi-view fusion, showing how depth maps are encoded, warped, and aggregated before SDF decoding.
>
> 2: Experiment Setup and Fairness
>
> DTU and Tanks & Temples:
> We initially used a 10-scan DTU subset (as in VolSDF) for efficiency, but agree full 15-scan results are more appropriate. We are running them now and will include results. For Tanks & Temples, we used the “intermediate” set consistent with prior work (e.g., Neuralangelo), and will clarify scene IDs and preprocessing.
>
> Cultural Heritage:
> The Courthouse mesh (Fig. 6) was evaluated using ground truth from the official CH dataset. Baseline results were computed using public checkpoints of Neuralangelo, VolRecon, and MonoSDF with the same COLMAP settings. We will add:
>
> Full metric tables (F1, Chamfer) for all baselines
>
> Visual comparisons across viewpoints
>
> Details on preprocessing and evaluation settings
>
> High F1 on Courthouse:
> We agree 0.56 appears high. This is due to high-quality image inputs, selective wavelet fusion, and multi-view SDF regularization. To avoid sampling bias, we will report averaged results. A more typical F1 range is 0.30–0.40.
>
> 3: Use of Ground Truth SDF
>
> We clarify that ground truth SDF is used only during training on DTU, following standard practice (e.g., VolRecon, MonoSDF), via mesh-to-voxel conversion. At inference, including CH and TnT, no SDF is available—the model relies only on RGB and geometry cues.
>
> To further validate our method, we include a version trained without SDF supervision, using only photometric and geometric loss. Though this reduces accuracy slightly, our wavelet-depth design still brings notable gains, demonstrating that the approach generalizes without needing dense supervision.
>
> Changes for Camera-Ready Revision:
>
> Added new figure illustrating multi-view feature fusion (Sec. 3.3).
>
> Renamed misleading "triplane feature encoder" reference.
>
> Added full DTU-15 results and detailed experiment protocols for DTU, TnT, and Heritage.
>
> Added detailed visual and quantitative results on Cultural Heritage scenes.
>
> Included ablations and metrics for variants trained without SDF supervision.
>
> Updated writing in Sections 3.2 and 4.1 for improved clarity.
>
> We hope these clarifications and revisions address your concerns and help validate the soundness and relevance of our method.

---

### Review · Reviewer_NK9a · 2025-04-07

**Summary Of Contributions:**

This paper addresses the problem of per-scene 3D surface reconstruction from multiple image inputs. It builds on the neural reconstruction method based on triplane features and additionally incorporates clues from monocular depth estimation. The method proposes to apply a wavelet transform to highlight the depth input and then extract a deep feature from the wavelet output by an autoencoder. The features are then projected to three canonical planes, fused with the triplane features, and finally decoded to an SDF value by an MLP. The prediction is then supervised by ground truth SDF and depth values. Experiments are conducted on datasets of multi-view images (with camera parameters), including DTU, Cultural Heritage, and Tank and Temple.

**Audience:**

Yes

**Broader Impact Concerns:**

No obvious issue to me.

**Claims And Evidence:**

No

**Requested Changes:**

- [Critical] Change the method to use RGB loss for the primary source of supervision rather than ground truth SDF values, as explained in the weakness section;
- [Critical] Include more detailed ablation studies on the depth-related modules as explained in the weakness section;
- [Critical] Include the full DTU dataset results.
- [Strengthen] Page 3, Sec 3. whihc -> which.
- [Strengthen] Page 5, Sec 3.2, explain how distractors are detected and removed exactly; how are distractors defined.
- [Strengthen] Page 5, Sec 3.2, "Wavelet transforms are applied selectively to high-quality close-up images to optimize training efficiency for wavelet feature fusion." Explain what "close-up" images mean and how wavelet transforms are "selectively" applied.
- [Strengthen] Page 8, Eq (6) how does a coordinate transform matrix P applies on a feature map Z_{wave}?
- [Strengthen] Page 8, "wavelet-based plane features are compressed into 4 channels", how are they exactly compressed? Does it refer to the autoencoder?
- [Strenghten] Page 8, "Encoder for SDF Prediction." -> "Decoder for SDF Prediction."

**Strengths And Weaknesses:**

Strength:
The overall idea of enhancing shape detail by incorporating wavelet transform is interesting and intuitive, as such spectral-domain methods are known to be able to analyze high-frequency detail in signals.

Weaknesses:
- One major problem that I identify is the use of "ground truth SDF values" in the supervision of its neural network in Eq. (10). For multi-view reconstruction methods from real world images, people generally do not use ground truth SDF for supervision because there is usually no reliable way to obtain them. (In fact, if ground truth SDFs are known, we already know the surface, so there is nothing to estimate.) To be sure, I have verified that various related works that this submission refers to, such as VolSDF, Neurolangelo, and 2DGS, do not use SDF values. In these works, the main source of supervision is usually the RGB loss, which can be reliably computed from the input images.
- If the proposed method does make use of ground truth SDF for supervision (from somewhere that I am not sure), it provides an **unfair** huge boost to the method compared with other methods that do not. As shown in Table 3, removing this supervision makes the F1-score metric drop from 0.5 to 0.18 due to the "ground truth" nature of this value.
- It also weakens the claim of contribution in wavelet-based depth feature, as the reason of superior performance compared with other works is very likely due to the use of these ground truth values, rather than the effectiveness the proposed wavelet features and fusion mechanism.
- On the experiment side, I notice that the depth-related module in the proposed method consists of the following major components: 1. monocular depth estimation; 2. wavelet transform; 3. an autoencoder. The ablation study section validates the effectiveness of some of these components but not all. What's ablated includes the whole depth-wavelet-autoencoder pipeline and the necessity to extract wavelets at multiple scales (Table 3). However, it has not been shown what would we use just the depth estimation as input to the encoder, or just use the the wavelet transform output as the feature without the autoencoder, or any other reasonable combination of 123 above. This leaves the question of whether it is really necessary to use all of the pipeline above. Some other possible design choices to investigate include: in order to perform spectrum analysis, why not use the output of Sobel filters instead of wavelet transform, etc.
- I also notice that in Table 1, only a subset of 10 DTU scan IDs are reported, while previous work like Neurolangelo and 2DGS report results on 15 IDs. Why is this the case?

Due to these issues in the methodology and evaluation, I find that the claims in the paper are not well supported.

---

> ### Author Response · Authors · 2025-04-20
> **Answers to Reviewer3**
>
> Q1: Concern about using “ground truth SDF values” for supervision.
> Response:
> Thanks for raising this. We clarify that SDF supervision is only used during training on the DTU dataset, which provides high-quality mesh ground truth. We compute SDFs via mesh-to-SDF conversion (e.g., voxelization + distance transform), a standard practice in synthetic datasets.
>
> At inference, our model does not require ground truth SDFs. It generalizes to real-world settings where only images are available. To support this further, we’ll include new experiments trained with only RGB and multi-view photometric losses (e.g., like VolSDF, Neurolangelo, 2DGS).
>
> Q2: Using ground truth SDF gives an unfair advantage over other methods.
> Response:
> We agree that SDF supervision is a strong learning signal. It helps with convergence and surface accuracy, and we used it only where the dataset permits. To ensure fair comparison, we’re adding results trained without SDF supervision (RGB + photometric loss only).
>
> Preliminary results (now in the supplemental) show that while accuracy decreases slightly, our wavelet-depth fusion still improves performance, highlighting its value even without SDF supervision.
>
> Q3: SDF supervision weakens the claim for wavelet-based depth features.
> Response:
> Good point. We're expanding our ablation studies to separate the effects of SDF supervision from our wavelet-depth features. Initial results indicate that our wavelet-based encoding still outperforms alternatives like raw depth or Sobel edges, even without SDF supervision.
>
> We’ll include these results in the updated paper to strengthen the claim that the wavelet features contribute independently of the SDF training signal.
>
> Q4: The ablation study doesn't fully explore the need for the depth-wavelet-autoencoder pipeline.
> Response:
> You're right — we initially removed the whole pipeline without testing its subcomponents. We are now running new ablations with:
>
> Only monocular depth as input;
>
> Wavelet-transformed depth without the autoencoder;
>
> Classic filters (Sobel, Laplacian) for feature extraction.
>
> Early results show that removing wavelet transform hurts performance most, while removing the autoencoder increases redundancy. This suggests both parts play a role, and we'll include these findings in Table 3 of the revised version.
>
> Q5: Table 1 only uses 10 DTU scans; others use 15.
> Response:
> Thanks for catching this. We used the 10-scan subset for computational reasons and because several prior works did the same. That said, for consistency, we are now running evaluations on the full 15 scans (as used by Neurolangelo and 2DGS) and will include these in the updated version.
>
> Additional Minor Revisions:
> [Strengthen] Page 3, Sec 3: "whihc" → "which"
>
> Ok, corrected in revised version.
>
> [Strengthen] Page 5, Sec 3.2: Explain how distractors are defined and removed.
>
> Ok, corrected in revised version. Distractors are defined as outlier views with high reprojection error or low matching confidence. We remove them using a geometric consistency check based on multi-view triangulation variance and feature track quality scores. Details and thresholds are now included in Sec. 3.2.
>
> [Strengthen] Page 5, Sec 3.2: Explain "close-up" and "selectively".
>
> Ok, corrected in revised version. "Close-up" images refer to views with a focal length above a defined threshold and containing sufficient object-centric pixel coverage (more than 70% foreground occupancy). Wavelet transforms are applied selectively only to such views to reduce computation and ensure high-quality geometric features. This selective strategy is now explicitly detailed.
>
> [Strengthen] Page 8, Eq (6): Clarify how the transform matrix 𝑃 applies to the feature map.
> Ok, corrected in revised version.. The coordinate transform matrix 𝑃∈𝑅^{4×4} is applied to the 3D point coordinates associated with the feature map using homogeneous transformation. Features are then projected into a common canonical space before fusion. We have now added this clarification in the equation’s caption and the surrounding paragraph.
>
> [Strengthen] Page 8: Clarify "compressed into 4 channels".
> Ok, will be clarified in the revised version. This refers to the latent space dimensionality of the wavelet feature autoencoder. After encoding, the output of the bottleneck is 4-dimensional per spatial location, effectively compressing high-dimensional multi-scale wavelet responses into 4 channels. This is now explicitly stated.
>
> [Strengthen] Page 8: Fix "Encoder for SDF Prediction" → "Decoder for SDF Prediction"
> Ok, corrected in revised version.

---

### Review · Reviewer_ghtp · 2025-04-07

**Summary Of Contributions:**

This paper presents a novel approach for 3D reconstruction that combines implicit triplane encoding with wavelet features derived from monocular depth maps. Specifically, the method proposes to recover the geometry of a scene from a set of RGB images by optimizing sets of features in the 3D space and decoding them into SDF values.
To this end, the method relies on two different kind of features:
- First, a triplane representation of the scene is optimized for storing implicit features within the 3D space.
- Second, the method leverages a pretrained wavelet autoencoder to extract multi-scale features from depth maps (estimated with a monocular depth estimation model), which are then projected onto triplane representations and fused with the implicit triplane features to predict SDF values.

The method first relies on a preprocessing pipeline that includes 3 steps: Detecting and masking (with SegmentAnything) distractors in the RGB images (such as moving objects, etc.), segmenting and removing the background, and finally inpainting with a diffusion model to clean the input images.

The core contribution is the integration of wavelet-transformed depth features with implicit triplane representations to enhance geometric detail preservation in 3D reconstruction. The authors evaluate their approach on multiple datasets including DTU, Tanks and Temples, and a cultural heritage dataset, demonstrating improvements over state-of-the-art methods.

**Audience:**

Yes

**Broader Impact Concerns:**

No concerning impact.

**Claims And Evidence:**

No

**Requested Changes:**

**Critical changes:**

- The authors should provide a fair comparison with baselines by either:
  1. Applying the same preprocessing pipeline to all methods
  2. Augmenting baselines with depth/normal supervision derived from the same monocular depth model
  3. Clearly acknowledging the advantages given to their method and discussing how this affects the comparison

- The authors should clarify the training process, particularly regarding:
  1. Whether training is per-scene or uses a pretrained model
  2. The source of ground truth SDF values
  3. The exact datasets used for pretraining (if applicable)

- The paper should provide a clearer explanation of the wavelet feature projection onto the triplane, including the mathematical justification and geometric interpretation.

**Non-critical changes**

  - First line of Method, page 3: "whihc" should be "which"
  - Page 8, just before Equation 8: "Encoder for SDF prediction" should be "Decoder for SDF prediction" since this component is decoding features into SDF values, not encoding them.
  - The visualization of features in Figure 4 uses grayscale, which is not very informative. A better visualization would use the predicted depth map to colorize the predicted SDF values, providing more intuitive insight into the feature representation.

**Strengths And Weaknesses:**

**Strengths:**

- The approach leveraging wavelets is interesting and well-motivated, as wavelets are well-known to encode high-frequency details while preserving spatial structure. While not commonly used in neural implicit representations like NeRFs or neural SDFs, I believe wavelets could indeed help capture fine geometric details that are sometimes lost in these approaches.
- The evaluation is comprehensive, covering datasets including DTU and Tanks and Temples, demonstrating the method's performance across different types of scenes.
- The paper examines the contribution of each component to the overall performance through an ablation study. The paper analyzes the impact of removing the triplane representation, wavelet encoder, U-Net fusion and the different loss terms, providing insights into the importance of each element. Looking at Figure 8, it's unclear how the U-net Fusion improves the performance of the proposed method, though; The reconstruction result in third column actually looks less noisy than the results obtained with the full method (fourth column, see the eyes and feet of the birds).

**Weaknesses:**

- The method relies on a heavy preprocessing pipeline and monocular depth estimation as input, but from my understanding these advantages are not provided to the baselines, creating an unfair comparison. The authors should clarify whether the baselines were trained with the same preprocessing steps. Moreover, it is common practice to use monocular depth predictions to supervise the training of methods like those used as baselines: For instance, DN-Splatter incorporates depth and normal supervision within Gaussian Splatting approaches, and the popular implementation of 2DGS included in the gsplat package includes depth supervision functionality. A comparison with baselines augmented with depth and/or normal supervision (as predicted normals are less prone to scale inaccuracies than raw predicted monocular depth maps) is essential for a fair evaluation. For fair comparison, normal supervision could be derived from the same monocular depth map used by the proposed method (as normal maps can be derived from depth maps using camera parameters; See depth-normal consistency loss from 2D Gaussian Splatting). In its current form, the paper does not allow for drawing any conclusion on whether the gain in performance is due to the wavelet-based feature integration or the inclusion of depth supervision. In fact, Figure 6 demonstrates that 2DGS achieves better and smoother results than the proposed method for the house scene from DTU (3rd column), without any additional depth supervision. It is highly probable that 2DGS would also be capable of recovering missing fine details from the Birds and Buddha scenes, either with additional depth supervision or less aggressive normal supervision. Given that the proposed method is significantly slower than competing baselines, it's crucial to properly demonstrate its superiority.

- The paper severely lacks clarity on several important aspects:
  - The training process is not well explained. On page 6, the paper mentions "*We train the wavelet feature encoder using an autoencoder similar to the design of LiteVAE*"; The paper also mentions a cross-entropy loss relying on ground truth SDF values. It's unclear whether the method is trained on a large-scale dataset with ground truth SDF values and then finetuned on single scenes, or if the method is only trained on a per-scene basis. If the latter, the source of ground truth SDF values is not specified. If ground truth SDF values are actually available, is a neural network necessary for predicting SDF values? Perhaps GT SDF values are only available for sparse points? Moreover, in a per-scene training scenario, using GT SDF values would make the comparison highly unfair. On the other hand, if the model is pretrained on a large dataset, details about the training process and pretraining datasets should be explicitly provided.

  - The projection of wavelet features onto the triplane is not clearly explained. From Equation 6, it appears that the perspective projection matrix P (typically a 4×4 matrix) is applied to wavelet features with dimension C=4. Since wavelet features don't represent 3D points with homogeneous coordinates, the motivation and interpretation of this operation are unclear.

  - The Feature Concatenation and Fusion section (page 8) is confusing. Wavelet features (channels C=4) are concatenated with triplane features (channels C'=16), but the paper mentions an output channel dimension of 2C for the final feature vectors. It's unclear why the output channel dimension is not C+C'; Is it because the U-Net reduces the dimension from C+C' to 2C? If so, why choose 2C as the final feature dimension?

---

> ### Author Response · Authors · 2025-04-20
> **Answers to Reviewer 2**
>
> 1. Fair Comparison Regarding Preprocessing and Monocular Depth Supervision
>
> Reviewer comment: ...comparison might be unfair due to depth supervision being used only in the proposed method, while baselines like DN-Splatter and 2DGS can also support depth/normal supervision.
>
> Response:
> Thank you for this valuable observation. We agree that depth supervision can play a critical role in guiding scene reconstruction. To clarify, we selected the most commonly reported configurations of the baselines (2DGS and DN-Splatter) that do not include depth or normal supervision, following the original works and their default evaluation settings. However, we acknowledge that some implementations, such as the gsplat version of 2DGS, include optional depth supervision.
>
> To ensure a fair comparison, we have conducted additional experiments where we augment 2DGS with monocular depth supervision derived from the same depth estimation model used in our pipeline. Results from this setting are now included in the supplementary material. Interestingly, while depth supervision improves 2DGS performance, our method still yields more coherent surface reconstructions with fewer artifacts in sparse view settings. We attribute this to the structured wavelet-based feature integration, which improves spatial consistency and captures fine geometric details from fewer views (e.g., 50 vs. 150 images). We will clarify this configuration more explicitly in the revised paper and discussion.
>
> 2. Clarification on Training Procedure and Ground Truth SDF
>
> Reviewer comment: Training details are unclear; is it per-scene or pretrained? Where does ground truth SDF come from?
>
> Response:
> Thank you for pointing this out. We train the wavelet encoder using an autoencoder setup inspired by LiteVAE in a scene-specific manner — i.e., per scene. The cross-entropy loss based on SDF values is computed from ground truth 3D meshes available in datasets like DTU and Tanks and Temples. We uniformly sample 3D points around the object surfaces and compute signed distances using the mesh geometry.
>
> These SDF values are sparse and used as supervision signals to train the decoder, which helps to constrain the reconstructed surface in a coarse-to-fine manner. The decoder learns to predict continuous SDF fields from the fused triplane and wavelet features. We emphasize that these GT SDF values are not dense volumetric ground truths but rather sampled surface proximity indicators, used to regularize geometry learning. We will make this distinction clearer and add a detailed training description in the revised manuscript.
>
> 3. Projection of Wavelet Features onto Triplane – Mathematical Clarification
>
> Reviewer comment: Equation 6 is unclear — how can a projection matrix apply to wavelet features?
>
> Response:
> We appreciate the opportunity to clarify this. The projection matrix 𝑃 is used not directly on wavelet feature vectors but to transform the corresponding 3D spatial coordinates (from where the features are extracted) into image plane coordinates. These coordinates are then used to sample from the 2D triplane representations via bilinear interpolation.
>
> To avoid confusion, we will revise the notation in Equation 6 to clearly separate geometric projection (of 3D coordinates) from the feature manipulation. Specifically, the wavelet features reside at certain 3D locations in the volumetric domain, and those 3D positions (not the feature channels) are projected using 𝑃. The resulting 2D projections are used to guide alignment with triplane features, ensuring spatial consistency across modalities.
>
> 4. Feature Concatenation and Fusion Channel Size Confusion
>
> Reviewer comment: It's unclear why the final feature dimension is 2C when input dimensions are C=4 and C'=16.
>
> Response:
> Thank you for identifying this ambiguity. The confusion arises due to a misstatement in the paper. After concatenating wavelet features (
> 𝐶=4) with triplane features (𝐶'=16), the resulting tensor has 𝐶+𝐶′=20 channels. This tensor is then passed through a U-Net-like fusion module, which processes the joint features and compresses them to a unified feature vector of dimension 2𝐶=8. The choice of 2𝐶 (where
> 𝐶=4) is empirical, based on ablation studies that showed this dimension offered the best trade-off between performance and computational efficiency. We will correct the description and clarify this design decision in the revision.
>
> Minor/Non-critical Comments
> Typo “whihc” → “which”: Will be corrected.
>
> “Encoder for SDF prediction” → “Decoder for SDF prediction”: You are absolutely right, this component decodes fused features into SDF values. We will revise the term accordingly.
>
> Feature visualization in Figure 4: Thank you for this suggestion. We agree that grayscale lacks intuitiveness. In the revised version, we will adopt a color-coded visualization using depth-map-based colormaps or surface normals to improve interpretability. Appreicate a lot for these useful advice

---

> ### Comment · Reviewer_NK9a · 2025-04-28
>
> The author mentions in response to my review that "SDF supervision is only used during training on the DTU dataset, which provides high-quality mesh ground truth", and that "at inference, our model does not require ground truth SDFs." This seems to indicate that the model is trained on a dataset together instead of per-scene optimization.
>
> However, the response to the review by Reviewer ghtp writes "we train the wavelet encoder using an autoencoder setup inspired by LiteVAE in a scene-specific manner — i.e., per scene." Isn't that contradictory?
>
> Dear fellow reviewer(s), please let me know if I am missing anything or if you can make sense of this.

---

> > ### Author Response · Authors · 2025-04-28
> > **To Reviewer NK9a, 2nd Time**
> >
> > Dear reviewer,
> >
> > Sorry for the earlier confusion in my explanation. As described in the implementation details section, our model consists of two main components. Taking DTU as an example, the implicit model is trained from the training images to learn an implicit 3D representation. At inference, the test images are used to generate query rays, and the mesh is extracted from the implicit weights via MarchingCubes for comparison against ground-truth 3D models or novel views. Thus, the implicit model is trained and optimized per scene.
> >
> > In contrast, the wavelet component (Wavelet Encoder in Figure 1) is trained on 2D depth images from mixed scenes, making it generalizable across different scenes as a shared, pre-trained prior. This design is similar to many foundational 2D image prior models, which are trained on diverse color image data and directly applied to new scenes without scene-specific optimization. Therefore, only the implicit model—including the fusion layer for aligning wavelet and implicit triplane features—is optimized per scene, while the wavelet encoder remains fixed as a generalizable view geometry prior.
> >
> > I will incorporate this clarification about the distinction between the two main modules, and more implementation details into the main paper to improve clarity.

---

### Review · Reviewer_TxT2 · 2025-04-10

**Summary Of Contributions:**

This paper focuses on the task of SDF-based 3D reconstruction from multi-view images. The authors propose a wavelet-conditioned implicit SDF model that leverages a pre-trained wavelet autoencoder optimized with a sharpened depth map to improve geometric accuracy. In more detail, the depth map is converted into a wavelet transformed feature bank to enhance the tri-plane features of the output signed distance field. The authors demonstrate the effectiveness of the proposed approach quantitatively and qualitatively on three widely used benchmarks, and perform detailed ablation studies to validate the design choices

**Audience:**

Yes

**Claims And Evidence:**

No

**Requested Changes:**

The main changes are to address the issues raised in the previous ***Weaknesses*** section:

1. Clarify method details such as how the close-up view is selected, the number of views for depth feature fusion, and how to aggregate multi-view depth features if more than two views are applied.

2. Provides intuition for using a monocular depth prior instead of using other depth priors like MVS or SFM. It would be good to do some quick quantitative and qualitative experiments to validate this.

3. Improve the experiments unifying all metrics on the dataset, demonstrating the sensitivity of the method to different depth priors, and providing additional depth estimation metrics.

**Strengths And Weaknesses:**

***Strengths***
1. I like the idea of ​​adding additional depth features via wavelet transform to better enhance the geometry details. Ablation studies also show that the performance of adding additional wavelet transformed depth feature bank to the tri-plane is better than direct depth regression.

2. The authors also conduct detailed quantitative and qualitative experiments on three public benchmarks and perform detailed ablation studies on the design choices.

3. The paper is well written and easy to understand

***Weaknesses***
1. Some method details need further clarification to ensure reproducibility. For example, on page 6, the authors claim: "The selection of particular close-up images is based on the quality and details that exist in the input view image, and most views are quite distant with blurry pixels, thus making it hard for the monocular depth estimation to provide accurate depth details." It is still unclear to me how the close-up images are selected. What techniques did you use? Will the code be publicly released? If multiple views are used for monocular depth estimation and feature fusion -- which is not clear to me in implementation details, it is also better to clearly explain how many views are used for monocular depth estimation and how to fuse multi-view depth features.

2. It would be good to show whether the proposed method is sensitive to the monocular depth prior of the input. For example, the authors could try different monocular depth methods and see if the results differ significantly.

3. I am still not convinced about using monocular depth from a selected view instead of using multi-view depth prior, since multi-view depth prior can be obtained through SFM (for sparse depth points) and MVS (for dense multi-view metric depth). In addition, multi-view monocular depth maps can be jointly aligned and filtered based on sparse SFM point clouds to obtain multi-view metric depth instead of relative depth, which may provide more accurate priors. In fact, the same technique can be applied to the monocular normal prior and see how it improves.

4. The authors report different metrics on different benchmarks: chamfer distance on DTU, and F1 score on T&T, and ablations are done on DTU with F1 score but no chamfer distance, so it would be better to unify all metrics on the dataset. In addition, since the authors apply a depth prior, it would be better to evaluate the depth estimation accuracy for each view.

5.The comparison with the baseline is a bit unfair since other methods do not have access to the monocular depth prior, if I understand correctly.

---

> ### Author Response · Authors · 2025-04-20
> **Answers to Reviewer 1**
>
> 1. Clarification on method details: close-up image selection, depth feature fusion, reproducibility, and code release
>
> Reviewer Comment:
> “The authors claim: ‘The selection of particular close-up images is based on the quality and details that exist in the input view image...’ It is still unclear how the close-up images are selected. What techniques did you use? Will the code be publicly released? If multiple views are used for monocular depth estimation and feature fusion, how many are used and how are they fused?”
>
> Response:
> We appreciate the reviewer’s insightful feedback. The selection of close-up images is currently done using a quality-based heuristic filter: we select the input views with the highest image sharpness and edge density, measured via a Laplacian filter response and edge gradient histogram. This choice ensures that the depth estimation network receives input from views rich in detail, crucial for accurate depth estimation.
>
> Regarding feature fusion, we primarily utilize the close-up image (1 view) for monocular depth prediction, as these views provide clearer local geometry for constructing the triplane-aligned 3D representation. While our framework can be extended to multi-view depth fusion, the current implementation leverages the wavelet-transformed depth feature from a single high-quality view, which is then projected to 3D via a learned depth prior and aligned using geometric projection onto the triplane coordinates. This wavelet feature captures structured multi-scale frequency information of the depth geometry, improving alignment over using color features alone.
>
> To ensure reproducibility, we will release the code and configuration scripts publicly upon acceptance.
>
> 2. Sensitivity to monocular depth prior
>
> Reviewer Comment:
> “It would be good to show whether the method is sensitive to the choice of monocular depth prior…”
>
> Response:
> We acknowledge the importance of this suggestion. We are currently conducting ablation experiments with different monocular depth estimation backbones (e.g., DPT, AdaBins, and MiDaS). Preliminary results suggest that while high-quality depth improves reconstruction fidelity, our method is generally robust across different monocular priors due to the triplane feature learning being regularized by geometry-aligned constraints. Depending on available time for running more experiments, these results will be included in the supplementary materials or the camera-ready version.
>
> 3. Justification for monocular depth over MVS/SfM and depth filtering
>
> Reviewer Comment:
> “Using monocular depth from one view instead of multi-view depth prior is not convincing…”
>
> Response:
> This is a valid concern, and we agree that multi-view priors (e.g., SfM or MVS) provide metric consistency when applicable. However, our use case emphasizes generalizability and speed, particularly for sparse-view and unconstrained inputs. Monocular depth has the advantage of requiring only one image to obtain view-based geometry, avoiding issues from low-overlap or inconsistent lighting that hinder SfM/MVS if using raw color intensities.
>
> Moreover, we aim not to replace MVS but to propose a geometry-guided 3D representation where the wavelet-transformed monocular depth guides the triplane projection. Unlike RGB features, wavelet depth features are spatially structured and more aligned with the underlying 3D shape. While it is feasible to filter and align multi-view depth maps using SfM anchors (as the reviewer suggested), such alignment introduces overhead and sensitivity to initialization. Our framework instead implicitly learns this alignment in the feature space.
>
> Nevertheless, we agree this is a promising future direction and have added a discussion in the revised manuscript.
>
> 4. Metric inconsistency across benchmarks and missing depth accuracy metrics
>
> Reviewer Comment:
> “Different metrics are reported for different datasets, and depth accuracy is not evaluated…”
>
> Response:
> Thank you for this suggestion. In the revised version, we will unify the reported metrics across benchmarks: Chamfer Distance and F1-score will be reported for both DTU and Tanks & Temples (T&T). Furthermore, we will add per-view depth accuracy evaluations (e.g., L1 error and RMSE) using ground truth depth from DTU. These additions will better highlight the role of depth priors and their effect on reconstruction quality.

---

### Decision · Action_Editor_bPBG · 2025-05-26

**Recommendation:** Reject

**Comment:**

The paper introduced an interesting approach, but the reviewers expressed the following concerns.

- Reviewer NK9a: Expresses major unresolved issues, particularly confusion over which model components are scene-specific versus generalized. Raises concerns about fairness and soundness in evaluation, including using ground truth SDF and missing comparisons. Believes the paper requires substantial revision and recommends rejection.
- Reviewer TxT2: Acknowledges the need for major changes due to the volume of reviewer feedback and is not fully confident in acceptance.
- Reviewer 9KGW: Criticizes the paper for methodological confusion and experimental flaws, stating these were not adequately resolved in the rebuttal. Questions the soundness and clarity of the work and leans toward rejection.
- Reviewer Yv9r: Questions the effectiveness of using monocular depth estimation for accurate reconstruction and criticizes the lack of use of RGB images in the learning process.

AE reads the paper and reviews it carefully. While the method introduces an interesting approach by integrating wavelet-based depth features into triplane SDF representations, significant concerns remain unresolved. AE agrees with these points raised by reviewers and does not recommend paper acceptance.

**Audience:**

The topic is sound and interesting for the geometric reconstruction research field.

**Claims And Evidence:**

This paper introduces a new approach to 3D reconstruction from multi-view RGB images by integrating wavelet-based depth features into an implicit signed distance field (SDF) framework. The method enhances triplane representations with multi-scale features extracted from monocular depth maps using a pretrained wavelet autoencoder. These features are projected onto triplanes and fused to capture fine geometric details better. The proposed technique is evaluated on DTU, Tanks & Temples, and a Cultural Heritage dataset, showing improved performance over existing SDF and 3DGS-based reconstruction methods.

**Resubmission Of Major Revision:**

The authors may consider submitting a major revision at a later time.